# ReTR: Modeling Rendering Via Transformer for Generalizable Neural Surface Reconstruction

**Yixun Liang**[1*]   **Hao He**[1,2*]   **Ying-Cong Chen**[1,2†]

[1] The Hong Kong University of Science and Technology (Guangzhou).
[2] The Hong Kong University of Science and Technology.
`yliang982@connect.hkust-gz.edu.cn, hheat@connect.ust.hk, yingcongchen@ust.hk`

## Abstract

Generalizable neural surface reconstruction techniques have attracted great attention in recent years. However, they encounter limitations of low confidence depth distribution and inaccurate surface reasoning due to the oversimplified volume rendering process employed. In this paper, we present Reconstruction TRansformer (ReTR), a novel framework that leverages the transformer architecture to redesign the rendering process, enabling complex render interaction modeling. It introduces a learnable *meta-ray token* and utilizes the cross-attention mechanism to simulate the interaction of rendering process with sampled points and render the observed color. Meanwhile, by operating within a high-dimensional feature space rather than the color space, ReTR mitigates sensitivity to projected colors in source views. Such improvements result in accurate surface assessment with high confidence. We demonstrate the effectiveness of our approach on various datasets, showcasing how our method outperforms the current state-of-the-art approaches in terms of reconstruction quality and generalization ability. *Our code is available at* `https://github.com/YixunLiang/ReTR`.

## 1 Introduction

In the realm of computer vision and graphics, extracting geometric information from multi-view images poses a significant challenge with far-reaching implications for various fields, including robotics, augmented reality, and virtual reality. As a popular approach to this problem, neural implicit reconstruction techniques [1, 2, 3, 4, 5, 6] are frequently employed, generating accurate and plausible geometry from multi-view images by utilizing volume rendering and neural implicit representations based on the Sign Distance Function (SDF) [7] and its variant. Despite their efficacy, these methods possess inherent limitations such as the lack of cross-scene generalization capabilities and the necessity for extensive computational resources for training them from scratch for each scene. Furthermore, these techniques heavily rely on a large number of input views.

Recent studies such as SparseNeuS [9] and VolRecon [8] have attempted to overcome these challenges by integrating prior image information with volume rendering methods, thereby achieving impressive cross-scene generalization capabilities while only requiring sparse views as input. Nevertheless, these solutions are still based on volume rendering, which poses some intrinsic drawbacks in surface reconstruction. Specifically, volume rendering is a simplification of the physical world and might not capture the full extent of its complexity. It models the interactions of incident photons and particles into density, which is predicted solely based on sampled point features. This oversimplified modeling fails to disentangle the contribution of the light transport effect and surface properties to the observed color, resulting in an inaccurate assessment of the actual surface. Moreover, the prediction of color

---

[*]Equal contribution.
[†]Corresponding author

37th Conference on Neural Information Processing Systems (NeurIPS 2023).

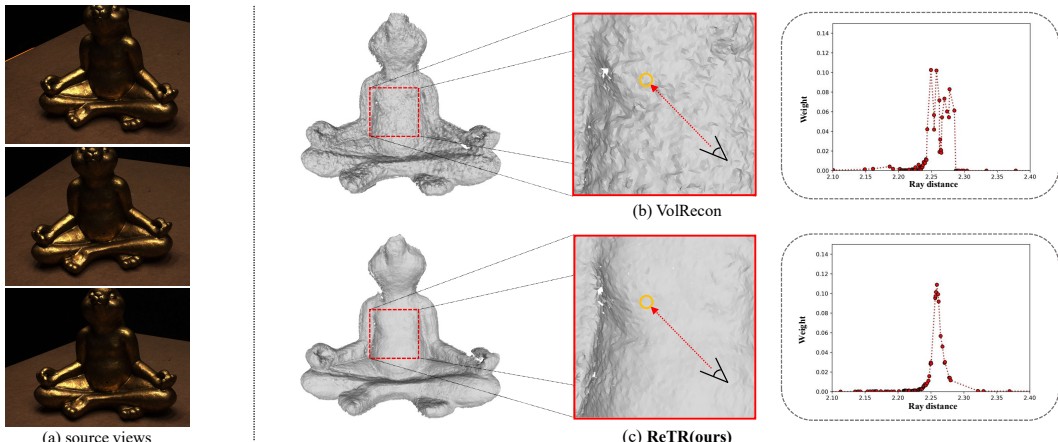

(a) source views      (b) VolRecon      (c) **ReTR(ours)**

Figure 1: Generalizable neural surface reconstructions from three input views in (a). VolRecon [8] produces depth distribution with low kurtosis and a noisy surface as shown in (b). In contrast, our proposed **ReTR** successfully extracts plausible surfaces with sharper depth distribution that has high kurtosis as shown in (c).

blending heavily relies on the projected color of the source views, thereby overlooking intricate physical effects. These shortcomings can lead to less confident surface predictions, producing a depth distribution with low kurtosis, and may be accompanied by high levels of noise, thereby compromising the overall quality of the reconstruction as illustrated in Fig. 1 (b).

In this paper, we first propose a more generalized formulation for volume rendering. Then based on this formulation, we introduce the Reconstruction TRansformer (ReTR), a novel approach for generalizable neural surface reconstruction. Our method utilizes the transformer to redesign the rendering process, allowing for accurate modeling of the complex render interaction while retaining the fundamental properties that make volume rendering effective. Particularly, we propose a learnable token, the *meta-ray token*, that encapsulates the complex light transport effect. By leveraging this token and the "cross-attention" mechanism, we simulate the interaction of rendering with the sampled points and aggregate the features of each point to render the observed color in an end-to-end manner. Additionally, we introduce a unidirectional transformer and continuous positional encoding to simulate photon-medium interaction, effectively considering occlusion and the interval of sample points. Moreover, as our model operates in the feature space rather than the color space, to further enhance the accuracy of semantic information and enable seamless adaptation, we propose a novel hybrid extractor designed to efficiently extract 3D-aware features.

Our method provides an excellent solution for generalizable neural surface reconstruction, offering several advantages over traditional volume rendering techniques. First, the ability to generalize complicated physical effects in a data-driven way provides an efficient approach to decoupling the contribution of light transport and surface to the observed color. Second, the entire process takes place in a high-dimensional feature space, rather than the color space, reducing the model's sensitivity to projected color in source views. Moreover, the transformer employs a re-weighted *softmax* function of the attention map, driving the learning of a depth distribution with positive kurtosis. As a result, our method achieves a more confident distribution, leading to reduced noise and improved quality, As shown in Fig. 1 (c).

In summary, our contribution can be summarized as:

- We identify the limitation and derive a general form of volume rendering. By leveraging this form, we can effectively tailor the rendering process for task-specific requirements.

- Through the derived general form, we propose ReTR, a learning-based rendering framework utilizing transformer architecture to model light transport. ReTR incorporates continuous positional encoding and leverages the hybrid feature extractor to enhance performance in generalizable neural surface reconstruction.

- Extensive experiments conducted on DTU, BlendedMVS, ETH3D, and Tanks & Temples datasets [10, 11, 12, 13] validate the efficacy and generalization ability of ReTR.

## 2 Related Works

**Multi-View Stereo (MVS).** Multi-view stereo methods is another branch for 3D reconstruction, which can be broadly categorized into three main branches: depth maps-based [14, 15, 16, 17, 18, 19], voxel grids-based [20, 21, 22], and point clouds-based [23, 24, 25]. Among these, depth maps-based methods are more flexible and hence more popular than the others. Depth maps-based methods typically decouple the problem into depth estimation and fusion, which has shown impressive performance with densely captured images. However, these methods exhibit limited robustness in situations with a shortage of images, thereby highlighting the problem we aim to address.

**Neural Surface Reconstruction.** With the advent of NeRF [26], there has been a paradigm shift towards using similar techniques for shape modeling, novel view synthesis, and multi-view 3D reconstruction. IDR [27] uses surface rendering to learn geometry from multi-view images, but it requires extra object masks. Several methods [2, 3, 1, 28, 29] have attempted to rewrite the density function in NeRF using SDF and its variants, successfully regressed plausible geometry. Among them, NeuS [2] first uses SDF value to model density in the volume rendering to learn neural implicit surface, offering a robust method for multi-view 3D reconstruction from 2D images. However, these methods require lengthy optimization to train each scene independently. Inspired by the recent success of generalizable novel view synthesis [30, 31, 32, 33, 34]. SparseNeuS [9] and VolRecon [8] achieve generalizable neural surface reconstruction using the information from the source images as the prior to neural surface reconstruction. However, these methods suffer from oversimplified modeling of light transport in traditional volume rendering and color blending, resulting in extracted geometries that fail to make a confident prediction of the surface, leading to distortion in the reconstructions. However, there are also some attempts [35, 36, 37] that adopt other rendering methods in implicit neural representation, such as ray tracing. However, these models are still based on enhanced explicit formulations; thus, their capacity to model real-world interaction is still limited. In contrast, our method introduces a learning-based rendering, providing an efficient way to overcome such limitations.

**Learning Based Rendering.** Unlike volume rendering, another line of work [38, 39, 40] explores deep learning techniques to simulate the rendering process. Especially Recurrent neural networks, which naturally fit the rendering process. Specifically, DeepVoxels [38] employs GRU to process voxel features along a ray, and its successor SRN [41], leverages LSTM for ray-marching. However, such methods recursively process features and demand large computation resources. The computation constraints inhibit such methods' ability to produce high-quality renderings. Unlike RNN-based methods, ReTR leverages the transformer to parallel compute each point's hitting probability. Greatly improve the efficiency of the rendering process and achieve high-quality renderings.

**Transformers With Radiance Field.** The attention mechanism in transformers [42] has also been widely used in the area of radiance field. In image-based rendering, IBRNet [30] proposes a ray transformer to process sampled point features and predict density. NeRFormer [43] utilizes a transformer to aggregate source views and construct feature volumes. NeuRays [44] leverages neural networks to model and address occlusions, enhancing the quality and accuracy of image-based rendering. GPBR [45] employs neural networks to transform and composite patches from source images, enabling versatile and realistic image synthesis across various scenes. However, these methods only use the transformer to enhance feature aggregation, and the sampled point features are still decoded into colors and densities and aggregated using traditional volume rendering, leading to unconfident surface prediction. Recently, GNT [46] naively replaces classical volume rendering with transformers in image-based rendering, which overlooks the absence of occlusion and positional awareness within the transformer architecture. In contrast to GNT, we improved the traditional transformer architecture in those two limitations based on an in-depth analysis of the fundamental components to make volume rendering work.

## 3 Methodology

In this section, we present an analysis of the limitations of existing generalizable neural surface reconstruction approaches that adopt volume rendering from NeRF [26] and propose ReTR, a novel architecture that leverages transformer to achieve learning-based rendering. We introduce the formulations of volume rendering and revisit its limitations in generalizable neural surface reconstruction in Sec 3.1. We then depict the general form of volume rendering in Sec. 3 and present

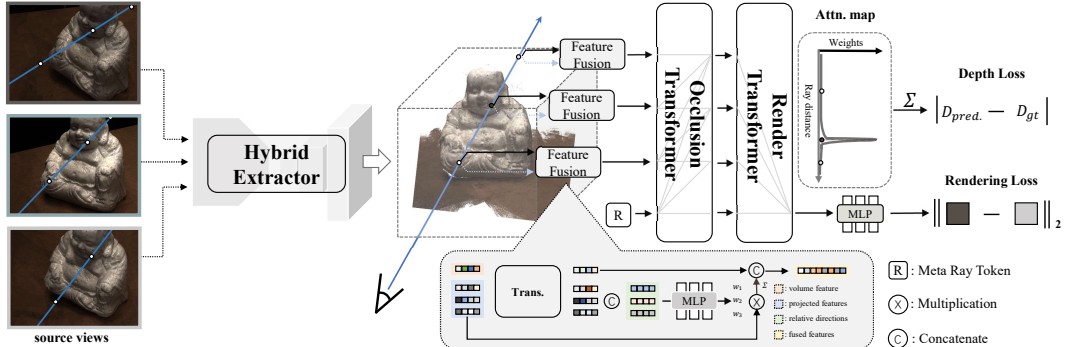

Figure 2: Our ReTR pipeline comprises several steps: (1). Extracting features through the proposed hybrid extractor model from source views, (2). Processing features in each sample point using the feature fusion block, and (3). Using the occlusion transformer and render transformer to aggregate features along the ray and predict colors and depths.

our learning-based rendering in Sec. 3.3. To effectively extract features for our proposed rendering, we further introduce the hybrid extractor in Sec. 3.4. Our loss functions are explained in Sec. 3.5.

## 3.1 Preliminary

Generalizable neural surface reconstruction aims to recover the geometry of a scene $\boldsymbol{S}$, which is represented by a set of $M$ posed input views $\boldsymbol{S} = \{\boldsymbol{I}_j, \boldsymbol{P}_j\}_{j=1}^{M}$, where $\boldsymbol{I}_j \in \mathbb{R}^{H \times W \times 3}$ and $\boldsymbol{P}_j \in \mathbb{R}^{3 \times 4}$ are the $j$-th view's image and camera parameters, respectively. Existing approaches [9, 8] generate the features of the radiance field from $\boldsymbol{S}$ using a neural network $\mathcal{F}_{enc.}$, and we formalize the process as:

$$\mathbf{f}^v, \{\mathbf{f}_1^{img}, \dots, \mathbf{f}_M^{img}\} = \mathcal{F}_{enc.}(\boldsymbol{S}), \tag{1}$$

where $\mathbf{f}^v \in \mathbb{R}^{R \times R \times R \times D}$ is the volume feature with resolution $R$ and $\mathbf{f}_j^{img} \in \mathbb{R}^{h \times w \times D}$ is the image feature with dimension $D$. To decode the color of a sampled ray $\mathbf{r} = (\mathbf{o}, \mathbf{d})$ passing through the scene, where $\mathbf{o}$ and $\mathbf{d}$ denotes as ray original and direction, the existing approaches [9, 8] sample $N$ points along the ray from coarse to fine sampling between near and far planes and obtain the location $\boldsymbol{x} \in \mathbb{R}^3$ of each sample point:

$$\boldsymbol{x}_i = \mathbf{o} + t_i \mathbf{d}, \quad i = 1, \dots, N. \tag{2}$$

The predicted SDF from features first converts to weights using the conversion function [2], denoted as $\sigma_i$. The weights are then used to accumulate the re-weighted projected colors along the ray using volume rendering. Specifically:

$$C(\mathbf{r}) = \sum_{i=1}^{N} T_i \left(1 - \exp\left(-\sigma_i\right)\right) \mathbf{c}_i, \quad \text{where} \quad T_i = \exp\left(-\sum_{j=1}^{i-1} \sigma_j\right), \tag{3}$$

$$\mathbf{c}_i = \sum_{j=1}^{M} \mathcal{F}_{weight.}(\mathbf{f}_i^v, \{\Pi(\mathbf{f}_k^{img}, \boldsymbol{x}_i)\}_{k=1}^{M})\Pi(\boldsymbol{I}_j, \boldsymbol{x}_i). \tag{4}$$

Here the $\mathcal{F}_{weight.}(\cdot)$ denotes the module that predicts the weight of each projected color. The $\mathbf{f}_i^v$ represents the volume feature at $\boldsymbol{x}_i$ obtained using trilinear interpolation in $\mathbf{f}^v$. $\Pi(\boldsymbol{I}, \boldsymbol{x})$ denotes the operation that projects $\boldsymbol{x}$ onto the corresponding input grid $\boldsymbol{I}$ and then extracts the feature at the projected location through bilinear interpolation.

**Limitation.** The volume rendering, denoted by Eq. (3), greatly simplifies light transport processes, thus introducing key limitations. A complete physical model for light transport categorizes the process into absorption, emission, out-scattering, and in-scattering. Each represents a nonlinear photon-particle interaction, with incident photon properties being influenced by both their inherent nature and medium characteristics. However, Eq. (3) condenses these complexities into a single density value, predicted merely based on sampled point features, leading to an *oversimplification*

*of incident photon modeling*. Moreover, the color determination method, influenced by a weighted blend of projected colors akin to Eq. (4), *over-relies on input view projected colors, overlooking intricate physical effect*s. As a result, the model requires a wider accumulation of projected colors from points near the exact surface, resulting in a "murky" surface appearance, as depicted in Fig. 1.

## 3.2 Generalized Rendering Function

To overcome the limitations we mentioned in Sec. 3.1, we propose an improved rendering equation that takes into consideration of incident photon modeling and feature-based color decoding. As we revisit Eq. (3) and determine its key components to facilitate our redesign. This differentiable equation consists of three parts: The $T_i$ term accounts for the accumulated transmittance and gives the first surface a bias to contribute more to the observed color. The $(1 - \exp(-\sigma_i))$ term denotes the alpha value of traditional alpha compositing, which represents the transparency of $x_i$ and is constrained to be non-negative. The color part of Eq. (3) denotes the color of $x_i$. Based on these analyses, we summarize three key rendering properties that the system must hold:

1. *Differentiable*. To enable effective learning, the weight function needs to be differentiable to the training network with observed color through back-propagation.

2. *Occlusion-aware*. In line with the bias towards the first surface in the original equation, the weight function needs to be aware that the points close to the first *exact surface* should have a larger contribution to the final output color than other points.

3. *Non-negative*. Echoing the non-negativity constraint in the alpha value, the weight of each point also needs to be positive.

Having identified these key properties, we can now reformulate our approach to meet these requirements. Specifically, we propose a more general form of Eq. (3), which can be formalized as:

$$C(\mathbf{r}) = \sum_{i=1}^{N} \mathcal{W}(\boldsymbol{F}_1, \dots, \boldsymbol{F}_i) \mathcal{C}(\boldsymbol{F}_i), \tag{5}$$

where $\boldsymbol{F}_i$ represents the set comprising image feature $f^{img}$ and volume feature $f^v$ in the $\boldsymbol{x}_i \in \mathbb{R}^3$, $\mathcal{W}(\cdot)$ is the weight function that satisfies those three key properties we mentioned above, and and $\mathcal{C}(\cdot)$ denotes the color function. Specifically, color $c$ can then be interpreted as characteristic of each feature point in 5. Therefore, feature at each point can be aggregated in a manner analogous to RGB value, and enabling us to deduce the primary feature points. This can be mathematically expressed as:

$$C(\mathbf{r}) = C(\sum_{i=1}^{N} \mathcal{W}(\boldsymbol{F}_1, \dots, \boldsymbol{F}_i) \boldsymbol{F}_i), \tag{6}$$

where the $C(\cdot)$ represents the color function that maps the feature into RGB space.

## 3.3 Reconstruction Transformer

Note that Eq. (3) can be considered as a special form of Eq. (5). With this generalized form, we can reformulate the rendering function to overcome the oversimplification weakness of Eq. (3). Based on Eq. (5), we introduce the Reconstruction Transformer (ReTR) that preserves essential rendering properties while incorporating sufficient complexity to implicitly learn and model intricate physical processes. ReTR is composed of a Render Transformer and an Occlusion Transformer. The Render Transformer leverages a learnable "meta-ray token" to encapsulate complex render properties, enhancing surface modeling. The Occlusion Transformer utilizes an attention mask to enable the occlusion-aware property. Also, ReTR works in the high-dimensional feature space instead of the color space, and thus allows for more complex and physically accurate light interactions. Consequently, ReTR not only overcomes the restrictions of Eq. (3) but also preserves its fundamental rendering characteristics. We elaborate on the design of these components as follows.

**Render Transformer.** Here, we discuss the design of the render transformer. Specifically, we introduce a global learnable token, refer to as "meta-ray token" and denote as $\mathbf{f}^{tok} \in \mathbb{R}^D$, to capture and store the complex render properties. For each sample ray, we first use the FeatureFusion block to combine all features associated with each sample point of the ray, resulting in $\mathbf{f}_i^f =$

FeatureFusion($\boldsymbol{F}_i$) $\in \mathbb{R}^D$. We then employ the cross-attention mechanism within the Render Transformer to simulate the interaction of sample points along the ray. It can be formalized as follows:

$$C(\mathbf{r}) = \mathcal{C}\left(\sum_{i=1}^{N} softmax\left(\frac{q(\mathbf{f}^{tok})k(\mathbf{f}_i^f)^\top}{\sqrt{D}}\right)v(\mathbf{f}_i^f)\right), \tag{7}$$

where $q(\cdot), k(\cdot), v(\cdot)$ denotes three linear layers and $\mathcal{C}(\cdot)$ in this formulation is an MLP structure to directly regress observed color from the aggregated feature, and $W(\cdot)$ translates to $softmax\left(\frac{q(\mathbf{f}^{tok})k(\mathbf{f}_i^f)^\top}{\sqrt{D}}\right)$. Furthermore, $C(\cdot)$ is operationalized as MLP, which serves to decode the integrated feature into its corresponding RGB value. Then, the hitting probability will be normalized by $softmax$ function, which is *Non-negative* and encourages the network to learn a weight distribution with positive kurtosis. And we can extract the attention map from the Render Transformer and derive the rendered depth as:

$$D(\mathbf{r}) = \sum_{i=1}^{N} \alpha_i t_i, \quad \text{where} \quad \alpha_i = softmax\left(\frac{q(\mathbf{f}^{tok})k(\mathbf{f}_i^f)^\top}{\sqrt{D}}\right), \tag{8}$$

where $\alpha_1, \ldots, \alpha_N$ denotes the attention map extracted from the Eq. (7). The rendered depth map can be further used to generate mesh [47] and point cloud [16].

**Occlusion Transformer.** To further make our system *Occlusion-aware* and enable of simulation of photon-medium interaction. We introduce Occlusion Transformer. Similar to previous works [46, 48], we introduce an attention mask to achieve that the sample point interacts only with the points in front of it and the meta-ray token. Such unidirectional processes encourage the later points to respond to the preceding surface. This process can be formalized as:

$$\mathbf{R}^f = \{\mathbf{f}^{tok}, \mathbf{f}_1^f, \mathbf{f}_2^f, \ldots, \mathbf{f}_N^f\}, \tag{9}$$

$$\mathbf{R}^{occ} = \text{OccTrans}(Q, K, V = \mathbf{R}_f),$$
$$\text{where} \quad \mathbf{f}_i^{occ} = \text{MLP}(\text{MHA}(Q = \mathbf{f}_i^f, K, V = \{\mathbf{f}^{tok}, \mathbf{f}_1^f, \ldots, \mathbf{f}_i^f\}) + \mathbf{f}_i^f. \tag{10}$$

Where MHA denotes the multi-head self-attention operation [42] and $\mathbf{f}_i^{occ}$ denotes the refine feature of $\boldsymbol{x}_i$ which obtained from the render transformer. In addition, $R^f$ is the collective set of tokens, and $R^{occ}$ signifies the occlusion transformer that employs $R^f$ for cross attention. Then, our Eq. (7) can be rewritten as:

$$C(\mathbf{r}) = \mathcal{C}\left(\sum_{i=1}^{N} softmax\left(\frac{q(\mathbf{f}^{tok})k(\mathbf{f}_i^{occ})^\top}{\sqrt{D}}\right)v(\mathbf{f}_i^f)\right). \tag{11}$$

**Continuous Positional Encoding.** Following the traditional transformer design, we need to introduce a positional encoding to make the whole structure positional-aware. However, positional encoding proposed in [42] ignores the *actual distance* between each token, which is unsuitable for our situation. Furthermore, weighted-based resampling [26] would lead to misalignment of positional encoding when an increased number of sample points are used.

To solve this problem, we extend the traditional positional encoding formula to continuous scenarios. Specifically, it can be formulated as follows:

$$PE_{(\boldsymbol{x}_i, 2i)} = sin(\beta t_i / 10000^{2i/D}),$$
$$PE_{(\boldsymbol{x}_i, 2i+1)} = cos(\beta t_i / 10000^{2i/D}). \tag{12}$$

Here, $i$ represents the positional encoding in the $i_{th}$ dimension and $\beta$ is a scale hyperparameter we empirically set to 100. The updated formula successfully solves the misalignment of traditional positional encoding, results are shown in Tab. 4. The specific proofs will be included in the Appendix section.

## 3.4 Hybrid Extractor

In our learning-based rendering, a finer level of visual feature is necessary, which is not achieved by traditional feature extractors that rely on high-level features obtained through FPN. These features

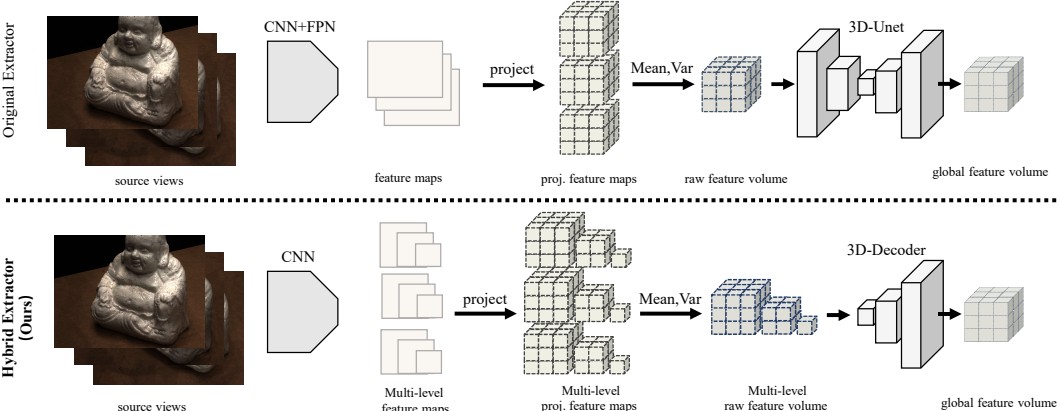

Figure 3: Comparision of the original extractor used in [8, 9] (top) and our **hybrid extractor** (bottom). The original extractor primarily discerns high-level features, demonstrating less efficacy. In contrast, our hybrid extractor excels in integrating multi-level features, demonstrating superior efficacy.

are highly semantic abstract and not suitable for low-level visual feature matching [49]. To overcome this limitation, inspired by NeuralRecon [50], we further propose Hybrid Extractor. Rather than relying on FPN to generate one feature map from high-level features of CNN, and using a 3D U-Net to process projected features as shown in Fig. 3, we leverage all level features from various layers to construct multi-level volume features. Then, we adopt a 3D CNN decoder to fuse and decode the multi-level volume features, producing the final global volume features.

Our approach enables us to perceive both low and high-level features, which is crucial for generalizable neural surface reconstructions that require detailed surface processing. Second, by avoiding the use of the encoder part of the 3D U-Net, we reduce the computational complexity and allow us to build a higher resolution volume feature within the same computational budget.

### 3.5 Loss Functions

Our overall loss function is defined as the weighted sum of two loss terms:

$$\mathcal{L} = \mathcal{L}_{\text{rendering}} + \alpha \mathcal{L}_{\text{depth}}, \tag{13}$$

where $\mathcal{L}_{\text{rendering}}$ constrains the observed colors to match the ground truth colors and is formulated as:

$$\mathcal{L}_{\text{rendering}} = \frac{1}{S} \sum_{s=1}^{S} \| C(\mathbf{r}) - C_g(\mathbf{r}) \|_2, \tag{14}$$

Here, $S$ is the number of sampled rays for training, and $C_g(\mathbf{r})$ represents the ground truth color of the sample ray $r$. The depth loss $\mathcal{L}_{\text{depth}}$ is defined as

$$\mathcal{L}_{\text{depth}} = \frac{1}{S_1} \sum_{s=1}^{S_1} | D(\mathbf{r}) - D_g(\mathbf{r}) |, \tag{15}$$

where $S_1$ is the number of pixels with valid depth and $D_g(\mathbf{r})$ is the ground truth depth. In our experiments, we set $\alpha = 1.0$.

## 4 Experiments

**Datasets.** The DTU dataset [10] is a large-scale indoor multi-view stereo dataset consisting of 124 different scenes captured under 7 different lighting conditions. To train our frameworks, we adopted the same approach as in previous works [9, 8]. Furthermore, we evaluated our models' generalization capabilities by testing them on three additional datasets: Tanks & Templates [12], ETH3D [13], and BlendedMVS [11], where no additional training was performed on the testing datasets.

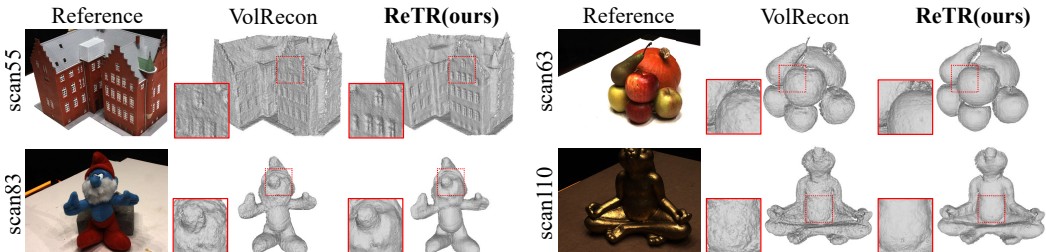

Figure 4: Sparse view reconstruction on testing scenes in the DTU [10]. Comparison with VolRecon [8] (left), our proposed ReTR (right) renders a more accurate surface and preserves finer details, *e.g.* the window of the house (scan 24), the nose of smurf, ReTR produces much sharper details. Best viewed on a screen when zoomed in.

| SCAN | Mean↓ | 24 | 37 | 40 | 55 | 63 | 65 | 69 | 83 | 97 | 105 | 106 | 110 | 114 | 118 | 122 |
|---|---|---|---|---|---|---|---|---|---|---|---|---|---|---|---|---|
| COLMAP [16] | 1.52 | **0.90** | 2.89 | 1.63 | 1.08 | 2.18 | 1.94 | 1.61 | 1.30 | 2.34 | 1.28 | 1.10 | 1.42 | 0.76 | 1.17 | 1.14 |
| MVSNet [51] | 1.22 | 1.05 | 2.52 | 1.71 | 1.04 | 1.45 | **1.52** | 0.88 | **1.29** | 1.38 | 1.05 | **0.91** | **0.66** | 0.61 | 1.08 | 1.16 |
| IDR [27] | 3.39 | 4.01 | 6.40 | 3.52 | 1.91 | 3.96 | 2.36 | 4.85 | 1.62 | 6.37 | 5.97 | 1.23 | 4.73 | 0.91 | 1.72 | 1.26 |
| VolSDF [1] | 3.41 | 4.03 | 4.21 | 6.12 | **0.91** | 8.24 | 1.73 | 2.74 | 1.82 | 5.14 | 3.09 | 2.08 | 4.81 | 0.60 | 3.51 | 2.18 |
| UNISURF [3] | 4.39 | 5.08 | 7.18 | 3.96 | 5.30 | 4.61 | 2.24 | 3.94 | 3.14 | 5.63 | 3.40 | 5.09 | 6.38 | 2.98 | 4.05 | 2.81 |
| NeuS [2] | 4.00 | 4.57 | 4.49 | 3.97 | 4.32 | 4.63 | 1.95 | 4.68 | 3.83 | 4.15 | 2.50 | 1.52 | 6.47 | 1.26 | 5.57 | 6.11 |
| PixelNeRF [32] | 6.18 | 5.13 | 8.07 | 5.85 | 4.40 | 7.11 | 4.64 | 5.68 | 6.76 | 9.05 | 6.11 | 3.95 | 5.92 | 6.26 | 6.89 | 6.93 |
| IBRNet [30] | 2.32 | 2.29 | 3.70 | 2.66 | 1.83 | 3.02 | 2.83 | 1.77 | 2.28 | 2.73 | 1.96 | 1.87 | 2.13 | 1.58 | 2.05 | 2.09 |
| MVSNeRF [31] | 2.09 | 1.96 | 3.27 | 2.54 | 1.93 | 2.57 | 2.71 | 1.82 | 1.72 | 2.29 | 1.75 | 1.72 | 1.47 | 1.29 | 2.09 | 2.26 |
| SparseNeuS [9] | 1.96 | 2.17 | 3.29 | 2.74 | 1.67 | 2.69 | 2.42 | 1.58 | 1.86 | 1.94 | 1.35 | 1.50 | 1.45 | 0.98 | 1.86 | 1.87 |
| VolRecon [8] | 1.38 | 1.20 | 2.59 | 1.56 | 1.08 | 1.43 | 1.92 | 1.11 | 1.48 | 1.42 | 1.05 | 1.19 | 1.38 | 0.74 | 1.23 | 1.27 |
| **ReTR (Ours)** | **1.17** | 1.05 | **2.31** | **1.44** | 0.98 | **1.18** | **1.52** | 0.88 | 1.35 | **1.30** | **0.87** | 1.07 | 0.77 | **0.59** | **1.05** | **1.12** |

Table 1: Quantitative results of **sparse view** reconstruction on 15 testing scenes of DTU dataset [10]. We report the chamfer distance, the lower the better, Methods are split into four categories from top to bottom: a) MVS-based methods, b) Per-scene optimization methods, c) Generalizable rendering methods, and d) Generalizable reconstruction methods. The best scores are in **bold** and the second best are in underlined.

**Baselines.** To demonstrate the effectiveness of our method from various perspectives, we compared it with (1) SparseNeus [9] and VolRecon [8], the state-of-the-art generalizable neural surface reconstruction method; (2) Generalizable neural rendering methods (3) Neural implicit reconstruction [27, 2, 1, 3] which require individual training for each scene from scratch. (4) Popular multi-view stereo (MVS) [16, 51] methods. Further details on the baselines are provided in the appendix.

## 4.1 Sparse View Reconstruction

For comparison, we performed sparse reconstruction using only three views, following the same approach as [8, 9]. We adopted the same evaluation process and testing split as used in previous works [8, 9] to ensure a fair comparison. We use a similar approach as VolRecon [8] to generate mesh, more details can be found in Appendix. As shown in Tab. 1, our method outperforms VolRecon [8] and SparseNeuS [9] by a significant margin. Moreover, our method also outperforms popular MVS methods such as MVSNet [51]. Furthermore, we present the qualitative results of sparse view reconstruction in Fig. 4. Our reconstructed geometry exhibits smoother surfaces and less noise compared to the current SoTA methods.

## 4.2 Depth Map Evaluation & Full View Reconstruction

We compare our rendered depth with those generated by SparseNeuS [9], MVSNet [51] and VolRecon [8]. Following the experiment settings introduced in VolRecon [8], we also use four source views as input for depth rendering. Additionally, we evaluated the performance by fusing all depth maps into a global point cloud. As shown in Tab. 2, our method outperforms existing methods in both evaluations. Moreover, as demonstrated in Fig. 5, our method achieves a sharper boundary with less noise and fewer holes compared to the current SoTA method [8].

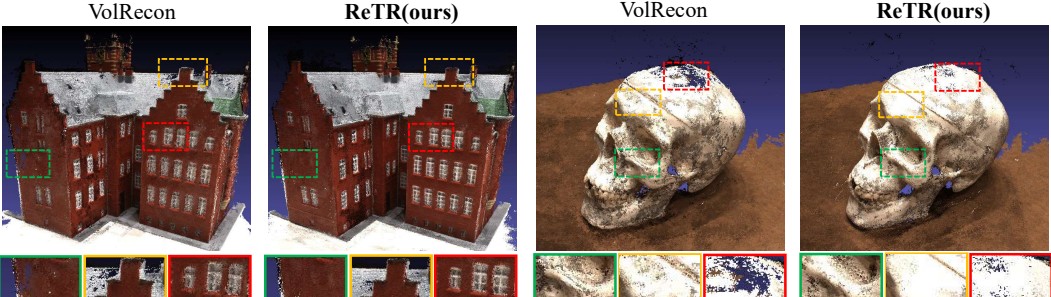

| VolRecon | ReTR(ours) | VolRecon | ReTR(ours) |

Figure 5: Full view reconstruction visualization in test set of DTU [10], comparison with VolRecon [8] (left), our proposed ReTR (right) reconstructs better point clouds, *e.g.* fewer holes, the skull head top, and the house roof gives a much complete representation, *e.g.* finer details, the house window, and the skull cheek, provides much finer details. Best viewed on a screen when zoomed in.

| Method | Acc. ↓ | Comp. ↓ | Chamfer ↓ | <1mm ↑ | <2mm ↑ | <4mm ↑ |
|---|---|---|---|---|---|---|
| MVSNet [51] | 0.55 | 0.59 | 0.57 | 29.95 | 52.82 | 72.33 |
| SparseNeuS [9] | 0.75 | 0.76 | 0.76 | 38.60 | 56.28 | 68.63 |
| VolRecon [8] | 0.55 | 0.66 | 0.60 | 44.22 | 65.62 | 80.19 |
| **ReTR (Ours)** | **0.54** | **0.51** | **0.52** | **45.00** | **66.43** | **81.52** |

Table 2: Quantitative results of **full view** reconstruction on 15 testing scenes of DTU dataset [10]. For the Accuracy (ACC), Completeness (COMP), and Chamfer Distance, the lower is the better. For depth map evaluation, threshold percentages (<1mm, <2mm, <4mm) are reported in percentage (%). The best scores are in **bold**.

## 4.3 Generalization

To evaluate the generalization ability of our model without retraining, we use three datasets, namely Tank & Temples, BlendedMVS, and ETH3D [12, 11, 13]. The high-quality reconstruction of large-scale scenes and small objects in different domains, as shown in Fig. 6, demonstrates the effectiveness of our method in terms of generalization capability.

## 5  Ablation Study

We conduct ablation studies to examine the effectiveness of each module in our design. The ablation study results are reported on sparse view reconstruction in test split following SparseNeuS [9] and VolRecon [8].

**Effectiveness of Modules.** We evaluate key components of our approach to generalizable neural surface reconstruction, as shown in Tab. 3. For the evaluation of the occlusion transformer, we keep the original transformer architecture while removing the special design we proposed in Sec. 3.3, to ensure the training parameter would not affect the evaluation. For the hybrid extractor part, we replace this module with the original extractor that has been used in [8]. Our results demonstrate that our approach can better aggregate features from different levels and use them more effectively. These evaluations highlight the importance of these components in our approach.

| Reder Trans. | Occ. Trans. | Hybrid Ext. | Chamfer↓ |
|---|---|---|---|
| ✓ | ✗ | ✗ | 1.31 |
| ✓ | ✗ | ✓ | 1.29 |
| ✓ | ✓ | ✗ | 1.28 |
| ✓ | ✓ | ✓ | **1.17** |

Table 3: Model component ablation. All of these parts are described in Sec. 3.3.

**Robustness of Different Sampling.** Tab. 4 displays the effects of altering the number of sample points on the reconstruction quality of VolRecon[8] and ReTR. Our method surpasses the current SoTA, even when the number of sampling points decreases. These results suggest that existing methods that rely on sampling points of the ray struggle to provide confident predictions of the surface due to the nature of volume rendering. Our approach, which uses learning-based rendering, is more resilient to sampling strategies and can provide reliable depth estimations even with fewer samples. Meanwhile, the effectiveness of continuous P.E. in Sec. 3.3 is proved through the result.

**Unsupervised Neural Surface Reconstruction.** Our approach is still applicable to unsupervised neural surface reconstruction using only colors for training, which remove $\mathcal{L}_{depth}$. Meanwhile, we find

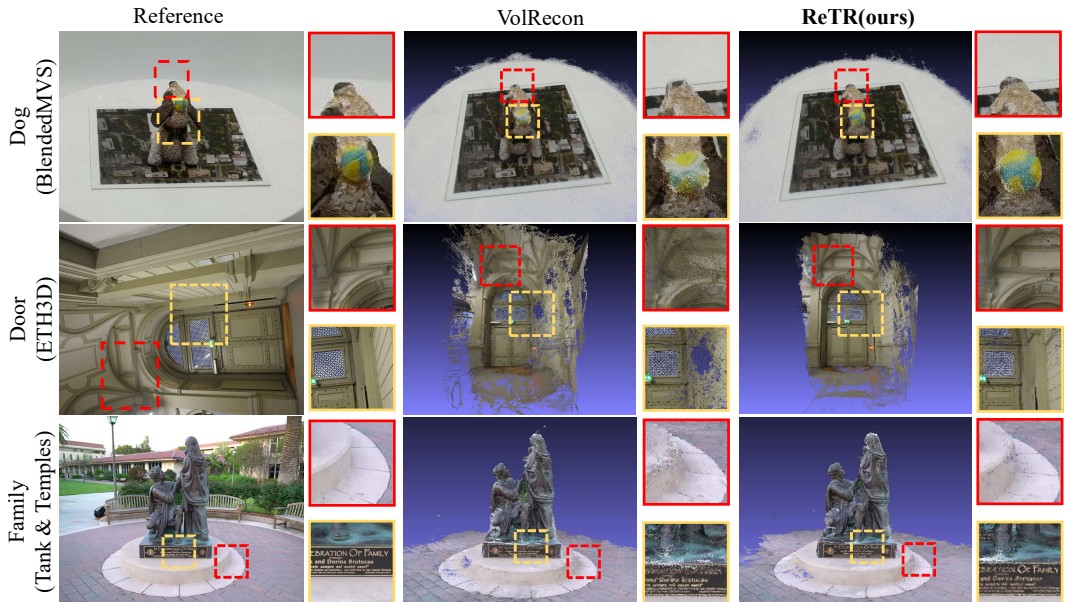

Figure 6: Generalize reconstruction visualization in 3 datasets BlendedMVS, ETH3D, and Tank & Temples [11, 10, 13], comparison with VolRecon [8] (middle), our proposed ReTR (right) generalize well to the large-scale datasets without fine-tuning. ReTR produces finer details and sharper boundaries. Best viewed on a screen when zoomed in.

| Sample Points | VolRecon [8] | ReTR (Ours*) | ReTR (Ours) |
|---|---|---|---|
| (16, 16) | 1.60 | 1.59 | 1.30 |
| (32, 32) | 1.45 | 1.42 | 1.20 |
| (64, 0) | 1.45 | 1.26 | 1.22 |
| (128, 0) | 1.79 | 1.46 | 1.20 |
| (64, 64) | 1.38 | 1.18 | **1.17** |

Table 4: Number of sampling ablation. The $*$ denotes our model using traditional positional encoding.

| Method | Training Losses | Chamfer ↓ |
|---|---|---|
| SparseNeuS [9] | w $\mathcal{L}_{depth}$ | 4.22 |
|  | w/o $\mathcal{L}_{depth}$ | 1.96 |
| VolRecon [8] | w $\mathcal{L}_{depth}$ | 1.38 |
|  | w/o $\mathcal{L}_{depth}$ | 2.06 |
| **ReTR (Ours)** | w $\mathcal{L}_{depth}$ | **1.17** |
|  | w/o $\mathcal{L}_{depth}$ | 1.45 |

Table 5: Ablation study of training losses.

that our method significantly outperforms the current SoTA method under unsupervised situations and is even comparable to COLMAP [16] the popular MVS technique, As shown in Tab. 5, This is further evidence that the improvement of complex rendering systems for implicit reconstruction is huge.

# 6    Conclusion

We have proposed Reconstruction Transformer (ReTR), a novel framework for generalizable neural surface reconstruction that uses transformers to model complex rendering processes. ReTR represents a significant advancement in the field of surface reconstruction, offering a powerful solution to the challenges faced by neural implicit reconstruction methods. Additionally, we delve into the design procedure of learning-based rendering. This exploration broadens our understanding of enhancing complex rendering systems and sets the stage for future research endeavors not only in surface reconstruction but also in other tasks relative to differentiable rendering.

# 7    Acknowledgements

We thank Shuai Yang and Wenhang Ge for the thoughtful review of our manuscript and valuable discussions throughout this project. Thank you to Yukang Chen, Shuhan Zhong and Jierun Chen for ideas and feedbacks on our manuscript. We would also like to thank the Turing AI Computing Cloud (TACC) [52] and HKUST iSING Lab for providing us with computation resources on their platform.

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

# Appendix

## A  Implementation Details

### A.1  Experimental Environment

Software and hardware environment:

- CUDA version: 11.1
- cuDNN version: 8.0.5
- PyTorch version: 1.10.1
- GPU: Nvidia RTX 3090
- CPU: Intel Xeon Platinum 8180 @ 2.50 GHz $\times$ 2

### A.2  Training Detail

Our model is implemented in PyTorch using the PyTorch Lightning framework. During the training stage, we resize the input image to 640 x 512 and the source views to $N = 4$. To train our model, we employ the Adam optimizer on a single Nvidia 3090 GPU. Initially, the learning rate is set to $10^{-4}$ and gradually decays to $10^{-6}$ using a cosine learning rate scheduler. Throughout the training, we use a batch size of 2 and set the number of rays to 1024. To enhance the sampling strategy, we apply a coarse-to-fine approach with both $N_{coarse}$ and $N_{fine}$ set to 64. The $N_{coarse}$ points are uniformly sampled between the near and far plane, while the $N_{fine}$ points are sampled using importance sampling based on the coarse probability estimation. Regarding the global feature volume $f^v$, we set its resolution to K=128. For inference on DTU, the image resolution is set to 800 x 600. For datasets such as BlendedMVS [11], ETH3D [13], and Tanks & Temples [12], we maintain the original image resolution. Training our model requires approximately 3 days on a single Nvidia 3090 GPU. Moreover, when constructing larger models such as the large and xlarge models by stacking more layers, the training time will naturally increase due to the increased model size.

### A.3  Mesh and Point Cloud Generation

Following the settings employed in VolRecon [8], we generate depth maps from virtual viewpoints by shifting the original camera along its $x$-axis by $d = 25$ mm. Subsequently, we perform TSDF fusion and applied the Marching Cubes algorithm to merge all the rendered depths into a voxel grid with a resolution of 1.5 mm and extract a mesh representation. For point cloud generation, we initially generate 49 depth maps by leveraging the four nearest source views. These 49 depth maps are then fused together to form a unified point cloud.

## B  Technical Details and Discussion

### B.1  Discussion of Hitting Probability

The attention score in ReTR can be interpreted as the probability of a ray being *hit*. However, when using *softmax*, the attention scores for each ray are forced to sum up to 1, implying that every ray should be considered a *hit*. To gain further insights, we examine the distribution of attention scores for rays that are *not hitting*. Figure 7 illustrates the results, demonstrating that the transformer intelligently employs a wider distribution to model rays that do *not hit*. The underlying rationale is that the transformer treats the surrounding air as a medium that contributes to the color.

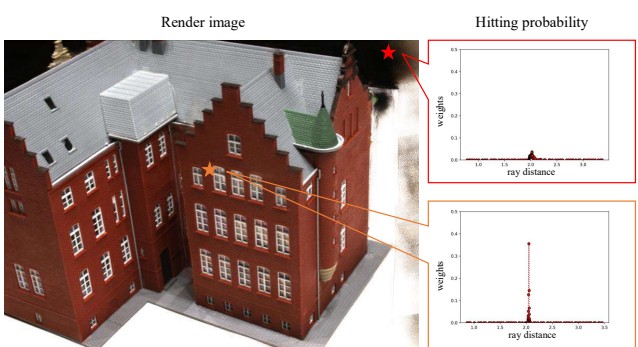

Figure 7: Hitting probability comparison.

| Sample Points | scan 24 | scan 37 | scan 40 | scan 55 | scan 63 | scan 65 | scan 69 | scan 83 | scan 97 | scan 105 | scan 106 | scan 110 | scan 114 | scan 118 | scan 122 |
|---|---|---|---|---|---|---|---|---|---|---|---|---|---|---|---|
| 16+16 | 1.09 | 2.49 | 1.51 | 1.09 | 1.62 | 1.64 | 0.97 | 1.35 | 1.43 | 1.05 | 1.21 | 0.77 | 0.72 | 1.27 | 1.28 |
| 32+32 | 1.06 | 2.30 | 1.46 | 0.97 | 1.35 | 1.53 | 0.89 | 1.38 | 1.34 | 0.92 | 1.10 | 0.74 | 0.60 | 1.10 | 1.17 |
| 64+0 | 1.11 | 2.39 | 1.43 | 1.06 | 1.36 | 1.62 | 0.94 | 1.28 | 1.31 | 0.91 | 1.12 | 0.78 | 0.64 | 1.18 | 1.20 |
| 128+0 | 1.39 | 2.36 | 1.54 | 1.01 | 1.18 | 1.65 | 0.97 | 1.26 | 1.26 | 0.83 | 1.10 | 0.84 | 0.62 | 1.09 | 1.15 |

Table 6: Chamfer distance of a number of different sampling points, results are shown for each scan under different settings.

When a ray does *not hit*, the transformer aggregates information from the surrounding air to obtain the color from these mediums.

## B.2 Hierarchical Volume Sampling through Attention Map

Given that our framework does not incorporate weights as seen in traditional frameworks like NeRF or NueS, we refine the original hierarchical sampling strategy by substituting the weights with the attention scores of each point. This approach, as discussed in the main text, is both straightforward and impactful. Additionally, we highlight that our method exhibits greater robustness in terms of the number of sampling points compared to the current state-of-the-art techniques, thereby offering an additional advantage within our framework.

## B.3 Continous Positional Encoding Proof

To imbue our system with positional awareness of actual distance. we initially derive the formula for the attention score, denotes as $s$ of features in $\boldsymbol{x}_i$ and $\boldsymbol{x}_j$:

$$s = (\mathbf{f}_i^f + \mathbf{p}_i)W_q W_k^\top (\mathbf{f}_j^f + \mathbf{p}_j)^\top, \tag{16}$$

where $\mathbf{p}$ represents the positional encoding in $\boldsymbol{x}$. We subsequently expand Eq. (16) as follows:

$$s = (\mathbf{f}_i^f)W_q W_k^\top (\mathbf{f}_j^f)^\top + (\mathbf{f}_i^f)W_q W_k^\top (\mathbf{p}_j)^\top + (\mathbf{p}_i)W_q W_k^\top (\mathbf{f}_j^f)^\top + (\mathbf{p}_i)W_q W_k^\top (\mathbf{p}_j)^\top, \tag{17}$$

where the fourth component of Eq.(17) denotes the interaction between two locations, and $W_q W_k^\top$ represents the trainable parameters. To ensure our MLP actual positional awareness, we need to make the function satisfy the following conditions:

$$(\mathbf{p}_i)(\mathbf{p}_j)^\top = f(t_j - t_i), \tag{18}$$

where $t_j - t_i$ denotes the distance between $\boldsymbol{x}_i$ and $\boldsymbol{x}_j$. When we apply our positional encoding (PE), the fourth component of Eq.(17) can be simplified as:

$$\begin{aligned}
(\mathbf{p}_i)(\mathbf{p}_j)^\top = \ & [sin(\beta t_i/10000^{2i/D}), cos(\beta t_i/10000^{2i/D})] \\
& \times [sin(\beta t_j/10000^{2i/D}), cos(\beta t_j/10000^{2i/D})]^\top,
\end{aligned} \tag{19}$$

$$\begin{aligned}
(\mathbf{p}_i)(\mathbf{p}_j)^\top = \ & sin(\beta t_i/10000^{2i/D})sin(\beta t_i/10000^{2i/D}) \\
& + cos(\beta t_j/10000^{2i/D})cos(\beta t_j/10000^{2i/D}).
\end{aligned} \tag{20}$$

By applying the sum-to-product identities, we obtain:

$$(\mathbf{p}_i)(\mathbf{p}_j)^\top = cos(\beta(t_j - t_i)/10000^{2i/D}), \tag{21}$$

where $(t_j - t_i)$ represents the actual distance between $\boldsymbol{x}_i$ and $\boldsymbol{x}_j$. Thus, continuous positional encoding enables the attainment of actual positional awareness.

## C  Additional Experimental Results

Here we show additional experiment results:

## C.1 Visualization Supplementary

Due to space limitations, we provide additional visual results for the experiments in this section. Specifically, we present the results for **sparse view reconstruction** in Fig. 9 and **full view reconstruction** of the point cloud in Fig. 11. Furthermore, we include the per-scene results for the number of sampling points in Tab. 6.

| Models | Mean | scan 24 | scan 37 | scan 40 | scan 55 | scan 63 | scan 65 | scan 69 | scan 83 | scan 97 | scan 105 | scan 106 | scan 110 | scan 114 | scan 118 | scan 122 |
|---|---|---|---|---|---|---|---|---|---|---|---|---|---|---|---|---|
| ReTR-B | 1.17 | 1.05 | 2.31 | 1.44 | 0.98 | 1.18 | 1.52 | 0.88 | 1.35 | 1.30 | 0.87 | 1.07 | 0.77 | 0.59 | 1.05 | 1.12 |
| ReTR-L | 1.16 | 0.98 | 2.26 | 1.59 | 1.00 | 1.14 | 1.56 | 0.90 | 1.35 | 1.26 | 0.86 | 1.06 | 0.78 | 0.57 | 1.01 | 1.07 |
| ReTR-XL | 1.15 | 0.96 | 2.26 | 1.64 | 0.94 | 1.19 | 1.59 | 0.86 | 1.32 | 1.25 | 0.85 | 1.02 | 0.75 | 0.55 | 1.02 | 1.11 |

Table 7: Result of ReTR-Base, ReTR-Large and ReTR-XLarge evaluated on DTU under 3 views setting. We report chamfer distance, the lower the better.

| Method | PSNR↑ | MSE↓ | SSIM↑ | LPIPS↓ |
|---|---|---|---|---|
| MVSNeRF* | 25.92 | 0.003 | **0.89** | **0.19** |
| VolRecon* | 23.37 | 0.004 | 0.80 | 0.30 |
| ReTR-B | 25.88 | 0.004 | 0.83 | 0.28 |
| ReTR-L | 26.03 | 0.003 | 0.84 | 0.27 |
| ReTR-XL | **26.33** | **0.003** | 0.84 | 0.27 |

Table 8: Novel View synthesis result on DTU, * denotes our reproduced result.

## C.2 Error Bar of ReTR

In order to assess the reproducibility and robustness of our model, we conduct three separate training runs using different random seeds. The corresponding results are presented in Figure 8. These results demonstrate that our model exhibits consistent performance across multiple training runs, indicating good reproducibility. Moreover, the minimal variance observes in the results further underscores the robustness of our model.

## C.3 Effectiveness of Stacking Transformer Blocks

To explore the potential of simulating more complex light transport effects, we extend our learnable rendering approach by stacking multiple layers of transformer blocks. Specifically, we introduce two variations: **ReTR-L**, where we stack two transformer blocks, and **ReTR-XL**, where we stack three transformer blocks. This allows us to experimentally evaluate the effectiveness of a more complex rendering system. The results of these experiments are summarized in Tab. 7. The findings indicate that by overlaying multiple layers of transformers, we can simulate complex lighting effects and achieve more powerful results. This demonstrates the potential of our approach to capture intricate light transport phenomena and enhance the overall rendering capabilities.

## C.4 Novel View Synthesis

In order to assess ReTR's performance in Novel View Synthesis, a task where many multi-view stereo techniques struggle, we conduct a quantitative comparison with VolRecon [8]. The novel views are generated during the full reconstruction for fusing the point clouds as we discussed in the main paper. Our results demonstrate a significant improvement over VolRecon in terms of novel view synthesis, as shown in Tab. 8. Additionally, we provide visualizations of novel view synthesis and depth synthesis in Figure 10. It has been challenging to achieve high-quality results simultaneously in rendering-based studies and reconstruction-based studies, with few methods excelling in both aspects. However, the results achieved by our proposed framework, ReTR, are highly promising, which suggests that a learnable rendering approach based on transformers can effectively integrate both tasks, yielding impressive results on both fronts within a unified framework.

# D Limitations

Our method requires approximately 30 seconds to render a depth map and image with a resolution of $600 \times 800$. Similar to other rendering-based methods such as IBRNet [30], VolRecon [8], and MVSNeRF [31], our approach has limitations in terms of efficiency. While learning-based rendering offers enhanced capabilities, it does introduce additional training parameters compared to traditional volume rendering techniques. Stacking multiple layers of our model can improve performance; however, it also increases training time due to the larger model size. It is important to strike a balance between achieving higher rendering quality and maintaining reasonable computational efficiency. Further research and optimization efforts can be explored to enhance the efficiency of our method,

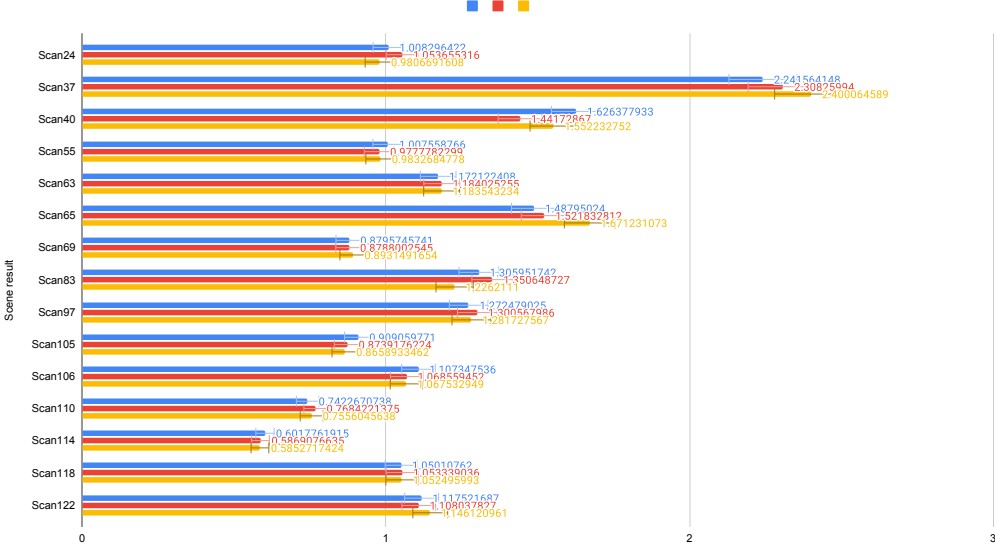

Figure 8: Error bar on 3 runs of Ours (ReTR).

potentially through techniques such as model compression, parallelization, or hardware acceleration. Acknowledging these limitations, we aim to provide a comprehensive understanding of the trade-offs between rendering quality, efficiency, and model complexity within our proposed framework.

## E  Broader Impacts

The proposed ReTR framework not only enables accurate surface reconstruction through learnable rendering but also generates high-quality novel views. These capabilities open up possibilities for various downstream applications in fields such as virtual reality (VR), robotics, and view synthesis with geometry. While these applications offer numerous benefits, it is important to acknowledge that they also come with ethical considerations. As authors of the ReTR framework, we are committed to promoting ethical practices and responsible development. We recognize the potential for misuse, such as generating content without consent, and we prioritize fair representation and responsible usage of the technology. We strive to adhere to ethical guidelines and contribute to the development of responsible AI practices. It is crucial to ensure that technological advancements are leveraged for the betterment of society while minimizing potential negative impacts. By maintaining a focus on ethics, fairness, and responsible development, we aim to ensure that ReTR and its applications are aligned with the principles of responsible AI and contribute positively to the broader scientific community and society as a whole.

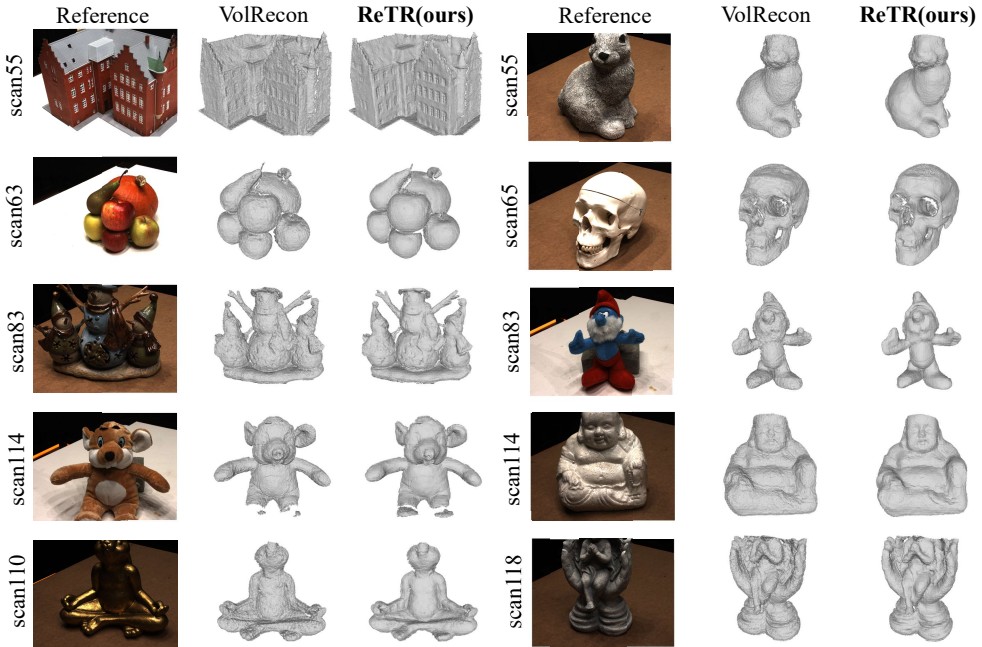

Figure 9: Comparison of VolRecon and ReTR in sparse view reconstruction with 3 input views.

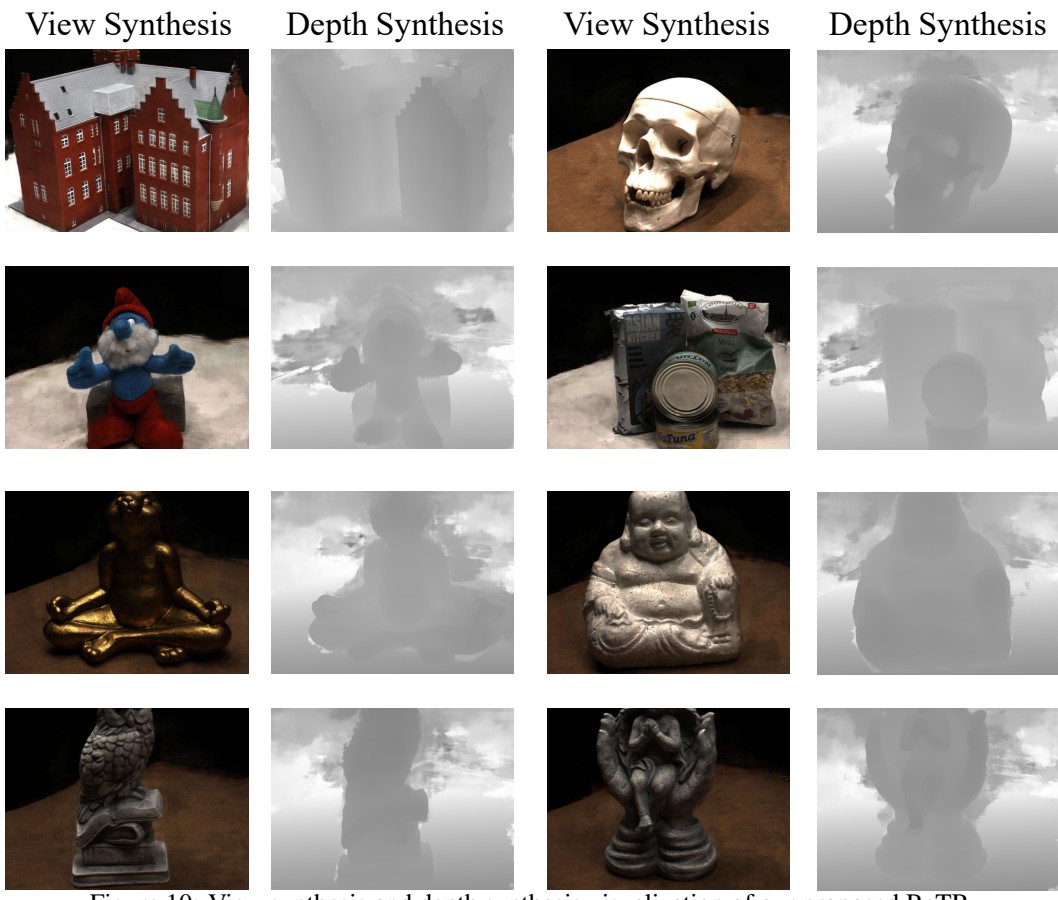

Figure 10: View synthesis and depth synthesis visualization of our proposed ReTR.

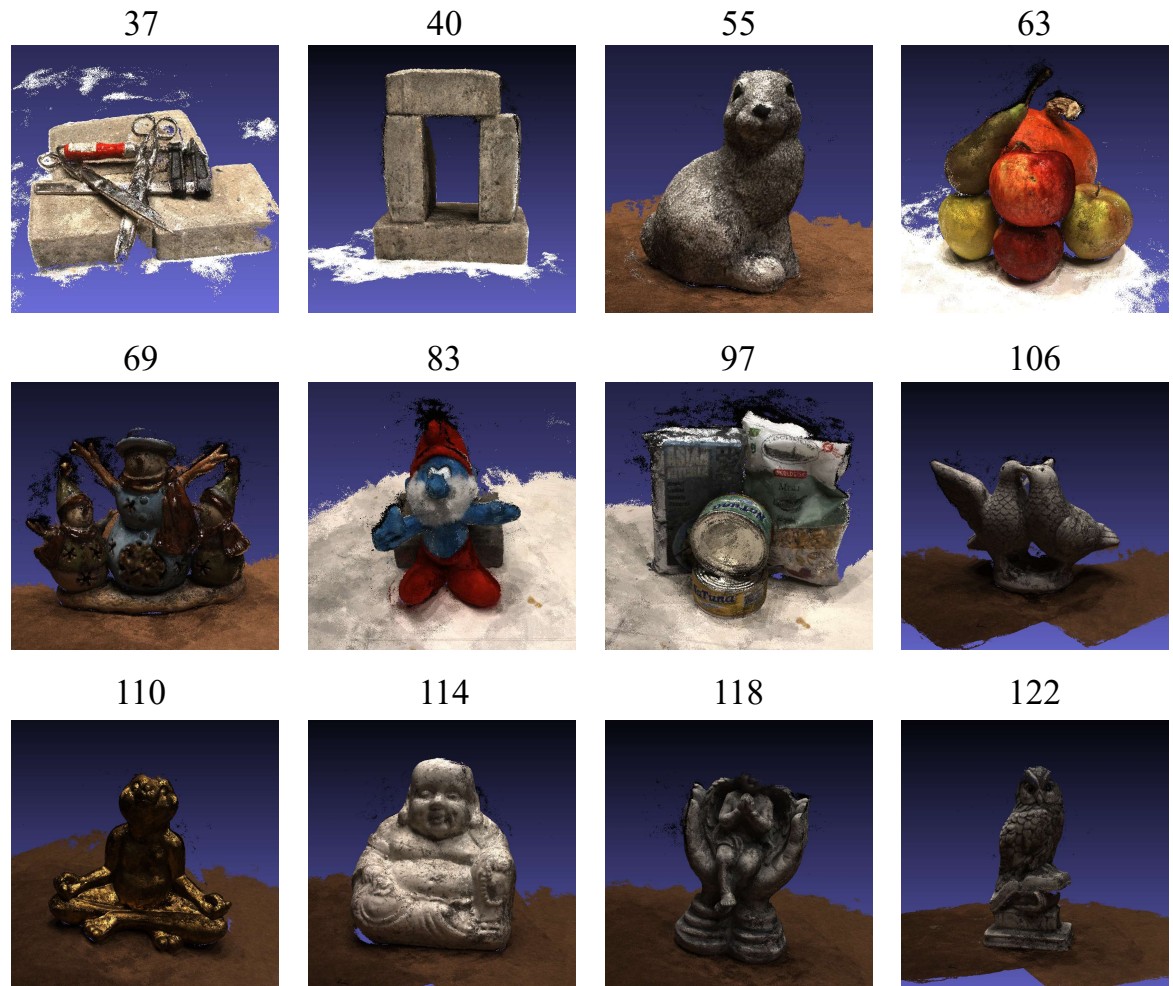

Figure 11: Visualization of full view generalization of a point cloud of our proposed ReTR.

