# OpenReview forum: "ReTR: Modeling Rendering Via Transformer for Generalizable Neural Surface Reconstruction"
_NeurIPS.cc/2023/Conference — NeurIPS 2023 poster_

### Official Review · Reviewer_1Qeo · 2023-06-22

**Soundness:** 3 good
**Presentation:** 2 fair
**Contribution:** 3 good
**Rating:** 6
**Confidence:** 5

**Summary:**

The paper presents a learning-based framework on the well-studied neural surface reconstruction problem. The key contribution of this paper is to take the complex photon-particle interaction into account and present a more generalized pipeline rather than relying on volume rendering. The proposed framework released the power of the transformer to achieve enhanced feature representation of sampled points along the ray. Experiments on several popular benchmarks have shown the effectiveness of the proposed approach.

**Strengths:**

(1) The idea of modeling the complex light transport and releasing the flexibility from regular volume rendering is novel and interesting.
(2) Overall, the paper is well presented and easy to follow.
(3) The paper has achieved state-of-the-art generalizable neural surface reconstruction performance across different datasets.


**Weaknesses:**

(1) (Major) The idea of using 3D feature volumes with hybrid resolution is not new and has been proposed in NeuralRecon (https://arxiv.org/pdf/2104.00681.pdf). Besides, since there are two major differences (the elimination of FPN and the construction of multi-level projected feature maps) between the proposed hybrid extraction and the original one, it is better to make separate ablation studies to further verify the effect of the two variations.
(2) (Major) For the ablation study of the occlusion transformer, compared with directly removing this module, a better ablation way is to attend every point’s feature as the input of the key embedding in the self-attention computation. This way ensures a fair setting (roughly same architecture and complexity) and the only difference is whether the later points contribute to the former ones.
(3) (Minor) Visual comparison on view synthesis with other methods: since one of the main motivations of this work is to model the photon-particle interaction, I assume a major outcome is more robust rendering against the variations (blur, specular..) from input views. Thus, it is better to show some visual comparison with other baselines (SparseNeus, VolRecon, NeRF…) to verify this point.
(4) (Minor)The main diagram (Figure 2) can be displayed in a clearer and more elegant way. Basically the part of transformer details would belong to either occlusion transformer or render transformer, rather than ‘feature fusion’. Besides, there is ambiguity on what the patches with different colors stand for.


**Questions:**

Overall I think the idea of the paper is novel and technically sound, presenting another perspective of the mainstream methods based on volume rendering. Despite some technical concerns listed in weaknesses, now I lean toward accepting this paper. I expect authors to address my concerns by providing more comprehensive experiments to further show the effectiveness of this work.

**Limitations:**

Authors have discussed their limitations on rendering speed. I think another limitation is that the SOTA neural surface reconstruction method is still only comparable with a classic MVS-based baseline (MVSNet) at this time. But MVSNet and its extensions must be much faster to get a depth map than the rendering-based ones. So there is a long way to go for this area to further release the power of implicit representation.

---

> ### Author Rebuttal · Authors · 2023-08-08
>
> We appreciate the reviewer’s time and insightful evaluations.
>
> **Q1. Hybrid resolution is not new, and justification of the proposed hybrid extractor with the original one.**
>
> Our main contribution is to rectify the oversimplification in traditional volume rendering by introducing a generalized form that better models incident photons. Yet HybridExtractor (HE) is a technical design that is advantageous in terms of efficiency and performance. We appreciate the reviewer highlighting the multi-level feature maps in NeuralRecon, and we will ensure to cite this paper and integrate discussion about this paper in the 3.4 section.
>
> To address the reviewer's concerns and showcase the effectiveness of HE in comparison with the baselines, we conducted additional experiments where HE was incorporated into the VolRecon framework. This was in response to the reviewer's observation regarding the "elimination of FPN and the construction of multi-level projected feature maps" as potential factors enhancing VolRecon's performance. Our experiments specifically evaluated the efficiency of both multi-level and single-level projected feature maps, resulting in a mean Chamfer distance of 1.35, marginally better than VolRecon's 1.38.
>
> At the core of HE is the ability to construct multi-level projected feature maps. In an attempt to address concerns about the exclusion of FPN, we developed a HybridExtractor variant that employs FPN for aggregating the features from the last two layers, resulting in a last-level projected feature. This model achieved a mean Chamfer distance of 1.34, indicating that the difference when not using FPN is minimal. However, it's worth noting that introducing FPN not only increases the parameter count but also enlarges the resolution of the last-level projected features, equating it with the second-level features. Our findings suggest that by constructing multi-level projected feature maps, we've essentially mirrored the multi-level feature aggregation function of the FPN, but in a more efficient manner.
>
> **Q2. Ablation study of Occlusion Transformer.**
>
> Thank you for the valuable suggestion. To clarify, the experiment was indeed conducted in a similar manner as suggested. Our occlusion transformer utilizes a mask, characterized by a diagonal matrix with its upper triangle masked out, ensuring that the subsequent point can only perceive points ahead of it. This design encourages the latter point to contribute to the object's surface formation. In the ablation experiment highlighted in $Tab. 3$ of the main document, we omitted this mask to observe any shifts in accuracy using chamfer distance as a metric. We acknowledge the confusion arising from the descriptions in $Tab. 3$ and lines 267-268 and will address this in our subsequent edition.
>
>  **Q3. Rendering against variations.**
>
> The goal of this work is to accurately reconstruct the surface geometry given sparse input views. However, ReTR still demonstrates strong performance in view synthesis. Here we show rendering results in rebuttal PDF Figure 3; for the “Train” scene, we can see ReTR is robust to the lighting changes: the background (red region in $Fig. 3$)  and the pavement (yellow region in $Fig. 3$) in front of the train, where VolRecon struggles to predict on such area. In addition, for “Scan 24” in DTU under different lighting, VolRecon again gives incorrect predictions about the front and back of the roof. Such a result shows ReTR is more robust under different settings.
>
> **Q4. To display Fig.2 in a more elegant way.**
>
> We will incorporate all the suggestions and revise them in the next version.

---

> > ### Comment · Reviewer_1Qeo · 2023-08-18
> >
> > Thanks for the valuable feedback from authors.
> >
> > Basically the feedback addressed most of my concerns. I expect the authors to have a better organized version in their final version on the language and diagram. Now I lean towards keep my original rating and accept this paper.

---

> > > ### Author Response · Authors · 2023-08-18
> > >
> > > Thank you for taking the time to provide your insights and for considering our work. We truly appreciate your feedback and will ensure that our final version will have improved language clarity and a better-organized diagram.
> > >
> > > Best,
> > >
> > > Authors

---

### Official Review · Reviewer_feWL · 2023-06-28

**Soundness:** 3 good
**Presentation:** 4 excellent
**Contribution:** 3 good
**Rating:** 7
**Confidence:** 5

**Summary:**

This paper proposes ReTR, a new architecture that leverages transformer to replace the traditional volume rendering process. The insight of the paper is that: the traditional volume rendering equation is oversimplified to model photon-particle interaction. Moreover, the color compositing function highly relies on the projected input view colors, and therefore overlooking intricate physical effects. To solve these two limitations, ReTR replaces the volume rendering equation with a render transformer. The attention map can be extracted from the render transformer to synthesize geometry details. An occlusion transformer is further introduced to obtain finer features. Instead of using FPN features and ResUNet, ReTR also utilizes features from different layers to construct multi-scale features. Experiments are conducted on the DTU dataset, Tanks & Temples, ETH3D, and BlendedMVS. The method is compared with state-of-the-art generalizable NeRF methods and generalizable surface reconstruction methods and achieves the best performance among them. Ablation studies also show the effectiveness of the network architecture. Moreover, ReTR surpasses SparseNeuS and VolRecon even without depth supervision.

**Strengths:**

I like the insights proposed in the paper that the traditional volume rendering equation is oversimplified. The solution that utilizes transformers to replace volume rendering is straightforward yet sound and effective. Experiments are exhaustive and validated the network designation. I also like the discussion of interpreting the render transformer the hitting probability.

**Weaknesses:**

More related work should be discussed in Section 2 for generalizable NeRF methods (NeuRays, CVPR22; Generalizable Patch-Based Neural Rendering, ECCV 2022, ...) and neural surface reconstruction methods (NeuS, ...).
Notations in Equations (8) and (9) are not well explained, for example, what are $\mathbf{R}^f$ and $\mathbf{R}^{\text{occ}}$, and $\mathbf{f}_i^{\text{occ}}$ did not appear before Equation (10).
There is also a typo at Line 232: **tanks and temples** instead of **tanks and templates**.

**Questions:**

Though the proposed method is effective, mathematically, I did not see how the network architecture can model the geometry better than other neural surface reconstruction methods, such as SparseNeuS and VolRecon -- Especially when the network is trained without depth supervision. I would like to see more insights and explanations from the authors in the feedback.

**Limitations:**

The main concern of this paper is it requires a long training time, e.g. 3 days on a 3090. It comes with a cost when introducing transformers in the network architecture. Generally, it is not a big problem since the method is generalizable.

---

> ### Author Rebuttal · Authors · 2023-08-08
>
> We appreciate the reviewer’s time and insightful evaluations.
>
> **Q1. More related work should be discussed in Section 2.**
>
> Thanks for the feedback. NeuRays(CVPR22) leverages neural networks to model and address occlusions, enhancing the quality and accuracy of image-based rendering. GPBR(ECCV22) employs neural networks to transform and composite patches from source images, enabling versatile and realistic image synthesis across various scenes. NeuS(NeurIPS21) uses SDF value to model density in the volume rendering to learn neural implicit surfaces, offering a robust method for multi-view 3D reconstruction from 2D images.  In the next version, we will discuss more relative works and incorporate all the suggested discussions into Section 2.
>
> **Q2. What are $R^f$ and $R^{occ}$, and $f^{occ}_{i}$ did not appear before Equation (10).**
>
> $R^f$ denotes the collective set of tokens. Specifically, $f^{tok}$ represents the "meta-ray token", employed to capture the global representation (as delineated in line 175). The remaining components, represented by $f_N$, describe the point features distributed along the ray.
>
> $R^{occ}$, on the other hand, signifies the occlusion transformer, which employs $R^f$ for cross attention.
>
> $f^{occ}_{i}$ denotes the refined features of $x_i$ which obtained from the render transformer.
>
> We recognize the importance of clarity and precision and will ensure that these notations are further elucidated and any typographical errors rectified in our forthcoming version.
>
> **Q3. How the network architecture can model the geometry better ... Especially when the network is trained without depth supervision.**
>
> Traditional volume rendering determines each point's hit probability based on its inherent features, with weights for each point then calculated via a set Cumulative Distribution Function (CDF). In this methodology, the impact of all prior points on the current point's weight is merely reflected in their cumulative probability.
>
> In contrast, our ReTR approach introduces a "meta-ray token" that serves as a global token, assimilating all features within a ray through cross-attention. This is similar to the **CLS_TOKEN** in **ViT**, wherein the global token will learn the statistical properties of rendering. The process then becomes about more than just individual point features; it's also influenced by the overarching information within the "meta-ray token," making for a richer information pool.
>
> Furthermore, our model trains using a rendering loss and is even effective **without** depth supervision. The softmax operation within our framework encourages the model to learn a specific peak – typically the surface – making it the primary influence on the rendered color. We conducted an experiment substituting Neus's rendering with ours (refer to the rebuttal PDF $Fig. 1$). The result indicated that our rendering approach guides the network towards a superior weight distribution (high kurtosis) along the ray, even when applying to per-scene optimization methods that do not require depth supervision, further showing the effectiveness of our proposed rendering approach. Traditional volume rendering can, on the other hand, produce flawed outcomes by identifying areas of low hit probability. Such outcomes are detrimental to accurate surface reconstruction, as we've illustrated in $Fig. 1$ of our main paper.

---

> > ### Comment · Reviewer_feWL · 2023-08-20
> > **Thanks for the rebuttal**
> >
> > Thanks to the authors for the rebuttal. All of my concerns are addressed. I decide to improve my rating to this paper.

---

### Official Review · Reviewer_RozF · 2023-07-01

**Soundness:** 3 good
**Presentation:** 3 good
**Contribution:** 3 good
**Rating:** 6
**Confidence:** 4

**Summary:**

The paper proposes a new framework for generalizable neural surface reconstruction, which utilizes the mechanism of transformers to model the rendering process. The authors first derive a general form of generalizable volume rendering based on existing methods and identify its limitations. They then suggest improving upon this framework by introducing a new rendering approach based on learned attention maps and performing over-ray accumulation in feature space rather than color space. Experiments are conducted on four different datasets and compared to recent baselines, demonstrating superior performance in generalizable reconstruction.

**Strengths:**

- The paper is well-written and exhibits a smooth flow. The authors effectively convey the motivation behind their methods, providing necessary background information and comparing against recent baselines. Additionally, the authors ensure that the reader can easily follow the logical progression of the paper.
- Section 3.1 presents a general framework that serves as a well grounded basis for existing methods. The authors' identification of limitations within this framework offers valuable observations, contributing to the overall storyline and motivation of the paper.
- The results presented in both the main paper and supplementary material demonstrate the superiority of the proposed method compared to the VolRecon baseline.
- The authors provided an ablation study on the various components of the method, effectively differentiating their individual contributions and providing a solid understanding of their impact.

**Weaknesses:**

- The section describing the reconstruction transformer appears to be incomplete and confusing due to several missing details and explanations:

  - It is unclear how equation 6 (and its improvement in equation 10) fit into the general framework proposed in equation 5. This confusion arises because the final MLP from feature to color does not align with the framework. Additionally, the weight function and color function are not explicitly provided.
  - The FeatureFusion operation is not defined, leaving ambiguity in understanding its purpose and implementation.
  - The definition of the "meta-ray token" $f^{tok}$ for a scene is unclear, specifically whether it pertains to per-image (per-ray/pixel) or per-scene information, and how it differs from the image features $f^{img}$.
  - The meaning of $F_i$ in line 161 is not provided or explained.

  It is crucial for the authors to address these misunderstandings and revisit the missing definitions in order to clarify the concepts.

- While it is acknowledged that the general form presented in section 3.1 oversimplifies the modeling of light transport in 3D scenes, it is hard to perceive how the mechanism of cross-attention over ray samples effectively models complex photon-particle interactions. Real interactions typically occur in spatial domains, whereas the suggested approach focuses on interactions over ray samples. It is suggested that the authors either temper these claims or provide further explanation on how their framework accounts for intricate global physical effects that encompass both global and local physical effects.

- The qualitative comparison is somewhat limited as it only includes a comparison to VolRecon. It is essential for the authors to provide visual comparisons with other methods as well, particularly SparseNeuS, to offer a more comprehensive evaluation.

- The authors have not presented timing evaluations of their method in both training and evaluation scenarios. Given that the limitation section highlights timing as a significant drawback of generalizable methods, it is important for the authors to address this by providing timing evaluations to enhance the paper's completeness.

**Questions:**

- When evaluating the SparseNeuS baseline, did the authors incorporate depth supervision as well? My understanding is that both ReTR and VolRecon utilize depth supervision during training, while SparseNeuS does not. This raises concerns about the fairness of the comparison between these methods.
- In Section 3.2, the paper suggests key rendering properties that the system should possess. However, there is no specific mention of the requirement for the weights to sum up to 1, indicating that all rays are eventually occluded. While this property is not explicitly described, it seems to be employed later in the paper using softmax. It would be helpful if the authors provided further clarification on this matter.

**Limitations:**

The authors have discussed limitations in the supplementary material. However, it is necessary to present the main limitation of efficiency in the main paper, even if briefly in the conclusion section. Additionally, providing quantitative results that demonstrate the tradeoff between the number of parameters and training/rendering time would greatly benefit the presentation of the method.

---

> ### Author Rebuttal · Authors · 2023-08-08
>
> We appreciate the reviewer’s time and insightful evaluations.
>
> **Q1. Reconstruction’s description appears incomplete and confusing.**
>
> **1.1. how does equation 6 (and its improvement in equation 10) fit into the general framework proposed in equation 5?**
>
> Thanks for the feedback. Since our rendering operates within the feature space, $Eq. 6$ naturally deviates slightly from $Eq. 5$. In the revised version, we elucidate these distinctions to ensure that readers can seamlessly navigate the transformations between the equations. Specifically, features will be aggregated using cross-attention (which can be analogized to the weight function in $Eq. 5$, with the attention map representing the weights). Subsequently, these aggregated features will be employed to predict RGB values (analogous to the color function in $Eq. 5$).
>
> Color can be interpreted as a characteristic of each feature point. In $Eq. 5$, we propose that the feature at each point can be aggregated in a manner analogous to RGB, enabling us to deduce the primary feature points. This can be mathematically expressed as:
>
> $C(\mathbf{r})=\mathcal{C}\left(\sum_{i=1}^{N}  \mathcal{W}\left({F}_1, \dots,{F}_i\right) {F}_i\right)$,
>
> Where the $\mathcal{C}\left( \cdot \right)$ represents the color function that maps the feature into RGB space.
>
> Building on this, we adapt the render transformer formulation in $Eq. 6$:
>
> $C(\mathbf{r})=\mathcal{C}\left(\sum_{i=1}^{N} softmax\left( \frac{q(\mathbf{f}^{tok})k(\mathbf{f}^{f}_i)^\top}{\sqrt{D}}\right)v(\mathbf{f}^{f}_i)\right)$,
>
> Where the $\mathcal{W}\left( \cdot \right)$ translates to $softmax\left( \frac{q(\mathbf{f}^{tok})k(\mathbf{f}^{f}_i)^\top}{\sqrt{D}}\right)$. Furthermore, in our approach, $\mathcal{C}\left( \cdot \right)$ is operationalized as an MLP, which serves to decode the integrated feature into its corresponding RGB value.
>
> We will incorporate this explanation into the next version.
>
> **1.2. Explanation of FeatureFusion Block.**
>
> The FeatureFusion Block is designed to merge both the volume feature and projected features at each point into a single integrated feature. This merged feature then undergoes subsequent operations. In detail, the volume and projected features are concatenated and passed through a transformer for refinement. The refined projected features, when combined with the relative direction of each image, undergo a MLP to infer the individual weights of the projected features. The outcome—a weighted sum of the projected features—is then concatenated with the refined volume features. This resultant feature serves as the foundation for the ensuing processes.
>
> **1.3. Explanation of "Meta-ray" token.**
>
> The "meta-ray token" acts as a universal token across the network, shared throughout scenes. At the outset of training, this token is initialized and engages in every rendering procedure during training, positioned as the primary token in the input sequence of sample points, akin to the CLS token in ViT. Distinctly, it does not derive from the point features of the input and remains separate from positional coding. As the network evolves continuously, this token gains the capability to encode specific statistical properties intrinsic to rendering.
>
> **1.4. Explanation of $f^{img}$ and $F_i$ in line 161.**
>
> $f^{img}$: This notation represents an intermediary phase of a feature from the input views, as extracted by our hybrid extractor. Later on, the conjunction of $f^{img}$ and $f^{v}$ amalgamate to create $f^f$. We recognize the need for clarity here and will address this in our upcoming version.
>
> $F_i$ in line 161: This notation represents the set comprising $f^{img}$ and $f^{v}$.
>
> We elucidate the above-mentioned issues in our revised version to alleviate any ambiguities.
>
> **Q2. Real interaction of light occurs in spatial domains… temper the claims.**
>
> We will tune down the claim the claims about particle interactions.
> Our primary contribution is the identification and rectification of inherent limitations in the widely used volume rendering approach, specifically its oversimplification of incident photon modeling. This issue has been largely unaddressed in prior works. However, modeling real interaction between photon particles face many challenges, such as the high cost of computation.
> We approximate this problem as interaction over ray samples, as we show in $Sec. 3$. In addition, our approach is much more cost-effective. We hope such observation can pave a new research direction for the community.
>
> **Q3. Qualitative comparison with SparseNeus.**
>
> Please kindly refer to the rebuttal PDF $Fig. 2$ for a comparative analysis between our proposed ReTR and SparseNeus. It's important to highlight that SparseNeus requires fine-tuning on specific scenes to attain the reported outcomes, whereas our approach involves direct inference on previously unseen scenes. We present results for both the direct inference and the fine-tuned scenarios for SparseNeus. In both instances, ReTR consistently demonstrates superior performance over SparseNeus.
>
> **Q4. Time evaluation.**
>
> Neus: RTX 2080ti, 16 hours per scene training.
>
> SparseNues: Two RTX 2080ti; pretraining takes 3 days and requires 20 mins per scene fine-tuning.
>
> VolRecon: single A100, pretraining takes 3 days; no further finetuning is needed.
>
> ReTR (ours): single RTX3090, pretraining takes 3 days; no further finetuning is needed.
> Inference takes about 30 seconds to render one image (DTU).
>
> **Q5. Depth supervision.**
>
> In $Tab. 5$ of the main manuscript, we compare ReTR against various baselines, considering both scenarios: with and without depth supervision. Notably, ReTR consistently surpasses these baselines in both settings.
>
> **Q6. Weight of the ray sum to 1.**
>
> For a more in-depth exploration of this issue, please refer to Appendix $B.1$.

---

> > ### Comment · Reviewer_RozF · 2023-08-19
> > **Post rebuttal**
> >
> > I want to thank the authors for making an effort in their rebuttal and addressing the reviewers' concerns.
> > The authors addressed most of my concerns. I still don't fully agree with the presented photon interaction over ray samples, and I feel like this discussion is a bit redundant.
> >
> > I lean on keeping my original score, since the paper requires additional clarifications. I suspect the paper requires a big revision to incorporate all the detailed explanations presented in the rebuttal.

---

### Official Review · Reviewer_TPgh · 2023-07-06

**Soundness:** 3 good
**Presentation:** 3 good
**Contribution:** 3 good
**Rating:** 5
**Confidence:** 3

**Summary:**

This works focus on generalizable asset reconstruction: given a few posed images, predict the 3D representations using a network.
Instead of using volume rendering to compute the transmittance, the authors propose to use transformer on the sampled points to compute the weight of each point.
Besides, the author also  improve the CNN architecture for feature extraction.
Extensive experiments are conducted on multiple datasets, demonstrating better performance.



**Strengths:**

1. Better performance compared to previous compared with previous SOTA methods
2. Code is attached and will be release.
3. method is well explained


**Weaknesses:**

1. Compared to previous methods SparseNeus and Volrecon, this work seems somewhat incremental. Major difference is using a hybrid CNN extractor (Fig 3) and a new transformer architecture. The "occlusion transformer" and "render transformer" seems to be two transformers with fancy names, and I don't see significant difference from the transformer in volrecon. Though Volrecon is a CVPR2023 paper, but apparently the authors use its codebase for develop , as can be seen in the code.zip in the supplementary.

2. I don't agree with the "rethink" title, equation5 doesn't make too much sense to me. It's not explained why the equation can satisfy the occlusion-aware, and especially no guarantee of consistency across multi views.
For example, for a sampled point $x$, its weight may  be $1$ for the ray $r_1$, but may be $0$ for the ray $C(r_2)$ even when $x$ is the nearest point in ray $r_2$. Therefore, I'm not fully convinced with that Eq 5 is a better modelling of the  rendering, and the "rethinking" seems be some sort of exaggeration.
Furthermore, given the goal of this work is reconstruction, I don't see why loosing the physics constrain in rendering can benefit the learned geometry.

## Justification of rating.
1. Pros: results are solid (multiple datasets, compared with SOTA baselines), code available.
2. Cons: Somewhat incremental, "rethinking" seems a bit exaggeration.

Overall I lean to a borderline accept, as no big technical flaws, but I'm not very confident.

**Questions:**

Please check the weakness section. Besides:

1. This may not be the weakness of this paper, but it's common in the research of this topic.  When it's called "generalizable", why it can not generalize to the unseen region of the object? For example, given images of the front views of the statue, why not generalize it to the backview in reconstruction.

2. What's the PSNR compared to other baselines

**Limitations:**

Discussion of limitation and broad impacts in the supplementary.
No license/asset description according to https://neurips.cc/public/guides/PaperChecklist . But I don't penalize it in the rating.

---

> ### Author Rebuttal · Authors · 2023-08-08
>
> We appreciate the reviewer’s time and insightful evaluations.
>
> **Q1. Incremental work of SparseNeus and Volrecon.**
>
> We respectively disagree with this. Our main contribution is to address the intrinsic limitation of the extensively-used volume rendering (line 25-37, $Fig. 1$ in the main paper), pointing out that it is an oversimplification of incident photon modeling (line 137-138), and it over-relies on input view projected colors (line 139-140). This is largely overlooked in previous works, including SparseNeuS and Volrecon. Then, we show that these limitations can be overcome by generalizing the volume rendering to the reconstruction transformer, which allows the modeling of complicated photon properties in the feature space. In contrast, VolRecon and SparseNeuS still rely on volume rendering that condenses complicated photon-particle interaction into a single density value. We also show that our approach significantly leads to more confident and accurate surface prediction both qualitatively ($Fig. 1$, $(c)$ in the main paper) and quantitatively ($Tab. 1$ in the main paper).
>
> **Q2.1 “Rethink” seems soft of exaggeration, and why loosing physics constrain benefits learned geometry.**
>
> We appreciate the feedback. The term "Rethink" in our context reflects a process of reevaluation and reconsideration of volume rendering. We observed the oversimplification issue in the current widely used rendering pipelines and were prompted to seek an alternative approach that is more suitable for the reconstruction pipelines.
>
> **Q2.2 “Why the equation can satisfy occlusion awareness and multi-view consistency.**
>
> The weighting term in $Eq. 5$ (line 160-161) can be further reformulated to $Eq. 10$ (line 196-197), as described in our main paper. The occlusion-aware process is done by the occlusion transformer that only allows the later points to interact with the points in front of it and the meta-ray token. Specifically, we applied an attention mask that masked out the top part above the diagonal. This is to make sure that for a given point $x$, it can only interact with the points in front of it in order to encourage the points to respond to the preceding surface. The multi-view consistency is done by conditioning the view features, similar to the previous MVS-based methods, such as VolRecon.
>
> **Q3. Why it can not generalize to the unseen region of the object?**
>
> Typically “generalizable” is defined as the ability of the model to predict an unseen scene as we follow this setting from previous works of literature (SparseNeus, Volrecon), but this is an interesting proposal. Traditionally, we approach this task from a **perceptual** standpoint. In this light, we solely generate surface geometry from three views, rigorously constraining the result with the ground truth, eliminating any element of randomness. However, reconceptualizing this task from a generative vantage point is intriguing. This would enable the network to "imagine" aspects previously unseen or uncharted. We believe this offers a promising avenue for exploration.
>
> **Q4. PSNR comparison.**
>
> The goal of this work is to accurately reconstruct the surface geometry given sparse input views. Like many previous studies, we use the chamfer distance to measure the accuracy of our reconstructed meshes. Although our main focus isn't on novel view synthesis, we've included this aspect in the Appendix for a comprehensive review. Please refer to the $Tab. 3$ (line 75-76) in the Appendix.C for detailed results. It's worth noting that our method outperforms VolRecon in the novel view synthesis. For additional qualitative results, please see the rebuttal PDF $Fig. 3$.

---

> > ### Author Response · Authors · 2023-08-21
> >
> > Thank you for your constructive feedback. We acknowledge your reservations about using the term "rethinking" in our title. In light of your and AC's feedback, we've decided to revise our title to "**ReTR: Modeling Rendering via Transformer for Generalizable Neural Surface Reconstruction**." We believe this better encapsulates the essence of our paper without overemphasizing the novelty.

---

### Official Review · Reviewer_DVpn · 2023-07-09

**Soundness:** 3 good
**Presentation:** 3 good
**Contribution:** 3 good
**Rating:** 5
**Confidence:** 4

**Summary:**

This paper introduces a interesting solution for volume renderings in generalizable neural surface reconstruction by leverage Transformers to predict depths and colors from feature volumes. The results on sparse view reconstruction prove its useness.

**Strengths:**

The authors identify the limitation and derive a general form of volume rendering and propose ReTR, a learning-based rendering framework utilizing transformer architecture to model light transport.
A hybrid feature extractor is also proposed for achieving better performance.

**Weaknesses:**

Can we replace the volume rendering in the optimization-based methods (e.g. NeuS / VolSDF) with the learned Transformer ? Or is the solution only works for generalizable neural surface reconstruction?

What is the performance in scene-level sparse view reconstruction, i.e., Replica/ScanNet.

Will HybridExtractor also works for volume rendering based methods (e.g. SparseNeuS / VolRecon) ?

Why Transformer? Will CNN/MLP also works for this design?

**Questions:**

See the weakness above.

**Limitations:**

See the weakness above.

---

> ### Author Rebuttal · Authors · 2023-08-08
>
> We appreciate the reviewer’s time and insightful evaluations.
>
> **Q1. If our learned transformer can replace volume rendering for optimization-based methods.**
>
> Yes, we've conducted qualitative experiments, as shown in Figure 1 of the rebuttal PDF, using optimization-based methodologies. Rather than traditional methods, we used spatial points and applied our learning-based rendering for scene optimization. Our method notably outperforms NeuS qualitatively (high kurtosis of weights along the ray), hinting that transformers might introduce a new direction in optimization-based reconstruction, similar to NeuS and VolSDF.
>
> **Q2.  Performance in scene-level sparse view reconstruction.**
>
> In addition to evaluating the DTU and BlendedMVS datasets, we further show the reconstruction results of our ReTR on scene-level datasets, specifically ETH3D and Tanks & Temples, following the same setting as VolRecon. The outcomes of these evaluations are visually represented in $Fig. 6$ of the main paper. More results will be included in the appendix in the next version.
>
> **Q3. Will HybridExtractor also work for volume rendering-based methods?**
>
> Yes, the HybridExtractor (HE) is adept at extracting both low-level to high-level features. Additionally, it potentially diminishes computational complexity by circumventing the use of the encoder segment of the 3D U-Net. To further substantiate our proposition, we integrated our HE into the VolRecon, resulting in a notable enhancement in performance, achieving a mean cd of 1.35.
>
> **Q4. Why Transformer?**
>
> Transformers have been extensively validated across diverse tasks owing to their exceptional capability in handling sequential feature interactions. The light transport effect can analogously be construed as the interaction between a photon and a particle. As we model the light interaction as rays, the transformer becomes a nature choice as it can model the sequence of points along rays effectively. In contrast, CNNs and MLPs appear to demand more tailored designs. Thus, we opted for the transformer to implicitly model the light transport effect. Nonetheless, we deeply appreciate the reviewer for highlighting this matter, suggesting an avenue worthy of future exploration.

---

> > ### Comment · Reviewer_DVpn · 2023-08-19
> > **Thanks for the rebuttal.**
> >
> > Thanks for the rebuttal, most of my concerns are addressed. I tend to keep my score.

---

> > > ### Author Response · Authors · 2023-08-19
> > >
> > > Thank you for acknowledging our efforts to address your concerns. We appreciate the time and expertise you've invested in reviewing our work.
> > >
> > > Best,
> > >
> > > Authors

---

### Author Rebuttal · Authors · 2023-08-09

We deeply appreciate the reviewers for their thoughtful feedback and time invested in evaluating our work. We're heartened by Reviewer DVpn's commendation of our solution as "interesting in generalizable neural surface reconstruction" and by Reviewer TPgh's acknowledgment that our "results are solid."  We are pleased that Reviewer Rozf found our paper “well-written” and “presents a general framework serves as basis for existing methods”. Moreover, we are encouraged by Reviewer feWL characterization of our work as "sound and effective", noting the "insights proposed in the paper". And by Reviewer 1Qeo's praise for our “novel and interesting” idea, complimenting that our paper is “well presented and easy to follow”. We will address each of the reviewers' additional comments in our subsequent responses. Thank you again for your invaluable feedback.

---

### Comment · Area_Chair_gQJD · 2023-08-20
**Relation to Deep Voxels**

The AC would like to ask how much "rethinking" this is, given Deep Voxels (https://arxiv.org/pdf/1812.01024) used a recurrent network (GRU) along the ray or Deep Shading (https://arxiv.org/abs/1603.06078) used a CNN to shade. Both "learn" the rendering.

Later, this fell in disfavour for volumes, and was replaced by differentiable non-learned components, popularized in NeRF.

Now this submission proposes transformers/attention, a concept more similar to GRU again than the VRE.

It feels like we move in circles. This  can be okay, but authors have to acknowledge similar previous attempts. So what is different to what Deep Voxels had to offer already 5 ys ago? If it is just, that a Transformer is a better way to learn ray processing than a GRU?

---

> ### Author Response · Authors · 2023-08-21
>
> We thank AC for highlighting previous similar works. In our subsequent version, we will incorporate a discussion on these studies.
>
> Specifically, AC brought attention to DeepVoxels[1], which employs GRU to process voxel features along a ray. We also noticed its successor, SRN[2], uses LSTM for ray-marching. We address these concurrently:
>
> While both transformers and GRU have similar algorithmic concepts, the way they handle and process sequence data sets them apart. Transformers are capable of parallel computation of sequence features, positioning them as particularly efficient for rendering tasks. One major advantage of volume rendering over earlier RNN-based methods is its efficiency in parallel computation of each point's hitting probability. This parallelism is crucial, as it allows for processing more points on the ray, leading to an increased resolution. Furthermore, evidence[3] suggests that the number of sample points directly impacts rendering quality. On the contrary, methods like DeepVoxels and SRNs recursively process features, which is resource-intensive. For instance, DeepVoxels demands 12GB VRAM for a $64^3$ voxel resolution, and SRNs, even with 4 Nvidia V100 GPUs, are capped at a 512x512 pixel resolution. Such constraints inhibit their capacity to produce high-grade renderings and sufficient training. As NeRF[3] discussed, SRN's view synthesis power is somewhat limited, emphasizing the advantages of modeling rendering as sample point probability distributions. In this context, Transformer is reminiscent of our approach that leverages attention to compute each point's hitting probability, drawing inspiration from volume rendering. Furthermore, like SRNs and DeepVoxels, our method can harness large data sets to understand rendering's statistical nuances. Our primary findings also suggest a synergy between these processes.
>
> In addition, AC also highlighted **Deep Shading**[4], a method that employs CNNs in screen space. Unlike the aforementioned methods, Deep Shading operates in a setting where the attributes of each pixel are provided prior. The 3D virtual scene's geometry and attributes are pre-known, enabling the network to "learn" shading based on screen space information. This approach maps various attributes to RGB values to achieve specific shading effects and improve rendering quality. On the other hand, methods like NeRF, Deep Voxel, and ReTR do not have this pre-existing scene information (i.e., the attributes present in Deep Shading). These methods employ different rendering techniques and leverage multiple 2D views provided to infer the surface of objects.
>
> Direct comparisons can be challenging given the diverse settings and objectives of methods like Deep Shading, DeepVoxels, and ReTR.
>
> We acknowledge the potential ambiguity of “Rethinking” in our initial title. The revised title will be: “**ReTR: Modeling Rendering via Transformer for Generalizable Neural Surface Reconstruction**.” We are grateful for AC's constructive feedback.
>
> If the AC has further concerns, please kindly let us know.
>
> **References**
>
> [1] DeepVoxels CVPR 2019
>
> [2] SRNs NeurIPS 2019
>
> [3] NeRF ECCV2020
>
> [4] Deep Shading CGF2017

---

### Decision · Program_Chairs · 2023-09-21

**Decision:**

Accept (poster)

**Comment:**

The paper proposes reconstruction with a learned image formation model based on Transformers. Previous models used physics-inspired equations instead.

The paper received lukewarm-to-supportive reviews. These were rebutted in detail, and for some resulted in a more positive angle. No agreement was achieved wether or not this new formulation is really beneficial or what the deeper inner logic is. The rebuttal did not really sway the mind of those questioning the idea. Authors also responded to the ACs questions in a plausible way that indicates they are able to change the paper.

Ultimately, the AC thought that the panel agrees that the paper's proposals of a learned image formation model for 3D reconstruction is 1) novel 2) demonstrated to improve the state of play. Most issues we see here come from presentation and language issues (no "photons" here, sorry) that could be fixed.